# Extensible Neck: A Gesture Input Method to Extend/Contract Neck Virtually in Video See-through AR Environment

**DOI:** 10.3390/s22093559

**Published:** 2022-05-07

**Authors:** Shinnosuke Yamazaki, Ayumi Ohnishi, Tsutomu Terada, Masahiko Tsukamoto

**Affiliations:** Graduate School of Engineering, Kobe University, 1-1 Rokkodaicho, Nada, Kobe 657-8501, Japan; shinnosuke-yamazaki@stu.kobe-u.ac.jp (S.Y.); ohnishi@eedept.kobe-u.ac.jp (A.O.); tuka@kobe-u.ac.jp (M.T.)

**Keywords:** human augmentation, gesture input, user interface

## Abstract

With the popularization of head-mounted displays (HMDs), many systems for human augmentation have been developed. This will increase the opportunities to use such systems in daily life. Therefore, the user interfaces for these systems must be designed to be intuitive and highly responsive. This paper proposes an intuitive input method that uses natural gestures as input cues for systems for human augmentation. We investigated the appropriate gestures for a system that expands the movements of the user’s viewpoint by extending and contracting the neck in a video see-through AR environment. We conducted an experiment to investigate natural gestures by observing the motions when a person wants to extend his/her neck. Furthermore, we determined the operation method for extending/contracting the neck and holding the position through additional experiments. Based on this investigation, we implemented a prototype of the proposed system in a VR environment. Note that we employed a VR environment since we could test our method in various situations, although our target environment is AR. We compared the operability of the proposed method and the handheld controller using our prototype. The results confirmed that the participants felt more immersed using our method, although the positioning speed using controller input was faster than that of our method.

## 1. Introduction

Many studies on systems for human augmentation virtually enhance the human body and capabilities using wearable devices. For example, Kasahara et al. proposed Jackln Head, which tracks the experience of another person remotely using an omnidirectional camera and HMD [1]. Nishida et al. developed a wearable visual translator that provides the perspective of a smaller person by shifting the wearer’s eyesight level down to their waist using a stereo camera and HMD [2]. With the advancement of human-augmentation technologies, we will use such systems for human augmentation in our daily lives.

We usually use handheld controllers to control various things such as video games, VR content, and drones. If such input devices are used to control functions in systems for human augmentation, the users can control these systems in the same way as controlling characters in video games. However, these devices are usually not intuitive for systems for human augmentation and require training sessions as well as controlling video games [3]. In our environmental assumptions, the users use systems for human augmentation anytime and anywhere. This means that these systems need an input method that reduces the cognitive load and can be operated intuitively and hands-free in daily life [4].

In this study, we focus on a human augmentation system that expands the movements of viewpoint such as extending and contracting the neck, as shown virtually in Figure 1. For several functions to control the virtual neck, we determine the appropriate motions for each, extending upward, forward, and backward, as shown in Figure 2. In our environmental assumption, the system for controlling the virtual neck consists of a video see-through HMD and extensible camera(s). Camera images are presented on the HMD, and the user controls the virtual neck by changing camera positions and angles. We also assume that people make a specific gesture when he/she imagines he/she wants to stretch the neck. For example, people naturally thrust their chin upward when they try to look beyond the wall taller than them. We consider these motions are also intuitive in systems for human augmentation while a human cannot extend their necks in real life. With the spread of HMDs and sensors, such unrealistic experiences are possible. However, to the best of our knowledge, no research has investigated what kind of motions are intuitive to people as input interfaces for such unrealistic augmentations.

In this paper, we investigated the natural gestures that provide input to this system by observing human motions when they want to extend/contract their necks. Furthermore, we determined the detailed operation methods including parameters for extending/contracting the neck and holding the position through experiments using “Tsumori control [5]”. This method is able to make an input as if he/she were controlling a particular object in motion, measure its sensor values, and use them as operating parameters. Then, we implemented a prototype of the human augmentation system to control the virtual neck. As a prototype, we construct a VR-based system to test various settings to investigate optimal input methods and evaluate our method in various conditions. Note that this study employed a VR environment to test our method in various situations, although this study’s target environment is AR.

The contributions of this paper are as follows:We propose that the input interface of a system that enables unrealistic human augmentation would be more immersive if the input method consisted of natural gestures that appear when the human augmentation is imagined and moved. Based on this hypothesis, we constructed a system and evaluated the input interface in this paper.We clarified the natural input gestures for a system that can expand the height of the viewpoint by extending and contracting the neck. These gestures were determined by observing the gestures which participants actually performed.We proposed a method of assigning the degree of chin thrust to a virtual neck length through an investigation using “Tsumori control”, in which behavioral intentions can be assigned to intuitive control methods.We implemented a prototype system and compared the operability between the motion-based operation and the handheld controller-based operation. The results showed that the controller input operation was faster in positioning. However, the motion-based operation was more immersive and augmented.

The remainder of this paper is structured as follows: In Section 2, we introduce related work. Section 3 describes the proposed method, and Section 4 describes the evaluation experiments. In Section 5, we discuss the limitations of this study. Finally, we summarize the paper in Section 6.

## 2. Related Work

Our work draws on four areas of research: human augmentation using HMDs, natural gestures as system input, Effects on the sense of physical ownership and immersion in virtual space, and Tsumori control. Relevant works on these topics are presented in this section.

### 2.1. Human Augmentation Using HMDs

Liang et al. proposed a field view augmentation that reduces the distortion of the omnidirectional camera images presented on HMD and enlarges the area of interest [6]. They found that the best method is to determine the user’s direction of interest by gazing for a period of time. The method based on the gaze reduced the user’s subjective motion sickness.

There are many studies on human visual augmentation using HMDs and Unmanned Aerial Vehicles (UAV). Okan et al. proposed a system that provides the users with X-ray vision using a UAV [7]. The user can explore the drone’s surroundings from the user’s perspective, even in a space with obstacles, by viewing rendered images from the UAV’s onboard camera through a see-through HMD. The UAV’s gesture-based operation methods include the pick-and-place and gaze-to-see functions. The pick-and-place function allows the user to move the UAV by gestures such as pinching it. The gaze-to-see function uses the user’s gaze acquired by the HMD to move the UAV to the area they are gazing at.

Higuchi et al.’s Flying Head synchronizes human head movements with a UAV equipped with a camera, and presents the UAV camera image to the HMD [8]. This system provides the user with the sensation that his/her head moves freely in the air. The user’s movements are mapped to the UAV’s movements using a motion capture camera. The camera limits the location because this system needs to capture the user’s movements from outside. In addition, the user can control the positions of the viewpoint higher than his/her by a handheld controller.

These related studies suggest that gesture-based system control is suitable for systems for human augmentation. Therefore, we conducted an observation to investigate intuitive input methods for human-augmentation system that can expand the movements of viewpoint by extending and contracting the virtual neck.

### 2.2. Natural Gestures as System Input

Rehman et al. proposed novel two hand gestures for three-dimensional navigation of a car in a VR environment [9]. This system recognizes the relative positions of the thumbs of both hands by acquiring the color marker attached to the user’s thumb from the camera attached to the HMD. The position determines various gestures similar to steering operation. The participants performed the task of driving on a road in the VR space using their method. The results showed that the method reduced their cognitive and learning load. Nakamura et al. proposed a method to manipulate the amount of displaying information in AR environments based on the amount of knotting between the eyebrows [10]. The manipulation by the eyebrows can be designed to respond to both unconscious expressions of emotions and conscious manipulation by the users. Ogawa et al. developed a virtual piano that can be played by dynamically changing virtual hands that differ greatly in appearance and movement [11]. In this way, the users can feel as if their own hands are transformed. Thus far, much research has used human gestures as input and operation methods for systems [12]. However, no research has investigated how to design intuitive operation methods for the unrealistic movement of extending/contracting the neck. In addition, a key to human-augmentation technologies is that they act like a natural extension of our own abilities, with no added effort or increased cognitive load required [4]. The natural gesture input reduces cognitive load.

We assume that the systems for human augmentation can be used anytime and anywhere in daily life. In this case, the user cannot always hold hand-controllers. We should use human motions as the input methods for the system. We determined the natural gesture for input methods by observing participants’ motions.

### 2.3. Effects on the Sense of Physical Ownership and Immersion in Virtual Space

In the VR space, avatars that move in sync with the user’s movements create a strong sense of body ownership [13]. Tabitha et al. investigated the change in implied ethnic bias in light-skinned participants by having them assume virtual bodies with different skin colors in a VR space [14]. The participants controlled a virtual body synchronized with their own movements in a VR space and checked their own appearances in a virtual mirror. This resulted in a higher sense of subjective ownership for the participants. In this study, we use a VR space that can simulate various environments, rather than the assumed AR environment. Therefore, it is necessary to make the VR space as close to the AR environment as possible. We introduce the phase into evaluation experiment to increase the participants’ sense of physical ownership of the avatar.

Immersion is important in games [15] and education [16,17] using VR and AR spaces since it is related to learning. Similarly, in body-augmentation systems using VR and AR, the sense of immersion and augmentation are considered to be important for reducing the cognitive load of the users and for learning. Sasaki et al. proposed an immersive virtual experience that provides the sensation of extreme jumping in the sky [18]. This system uses a propeller unit that generates wind in one direction depending on the elevation of the user’s viewpoint. This mechanism enhances the sense of immersion. Tachi et al. proposed a surrogate robotic system operating in a remote environment [19]. The system synchronizes the user’s visual, auditory, tactile and motor senses with the avatar. This synchronization enhances the user’s sense of physical ownership of the avatar. We aim to implement this system in the real world. Thus, we try to enhance the sense of immersion by making the user hear a sound that gives the impression of an object stretching when the virtual neck extends and contracts. Its sound helps the users to feel the extension of the neck not only visually but also audibly.

Thammathip et al. showed that the participants perceive themselves as superheroes when the interpupillary distance remains the same and the eye height increases in the VR space, and they perceive themselves as giants when the interpupillary distance increases and the eye height increases [20]. We expect that the sensation of extending and contracting the neck can be obtained by expanding the viewpoint shift through movements around the neck.

### 2.4. Tsumori Control

Miyamoto et al. proposed “Tsumori control” as one of the methods to intuitively operate multi-degree- of-freedom robots [5]. This control method enables intuitive operation by extracting and estimating the human’s discrete motion intentions from the continuous and intuitive motion output. They conducted an intention extraction experiment. They prepared a robot that performed a series of movements at regular intervals. The participants watched the sequence of motions and made inputs by exerting force on the control stick as if they were operating the robot themselves.

In this paper, we investigate to set up operating parameters such as neck extension/contraction speed and a detailed operation method to stop neck extension/contraction. Then, from the obtained sensor values, the operating parameters and operating methods are determined for each user. This method is also used for calibration before the evaluation experiment.

## 3. System Design

In this section, we describe the design of a human-augmentation system that can freely extend and contract the virtual neck. We assume that this system is used in daily life. For this reason, we need to consider intuitive gestures as input methods. However, this is an unrealistic motion because we cannot actually extend and contract our necks. Therefore, we set up three situations in which we wanted to extend the neck, and observed the motion of the participants. The investigation led us to determine gesture input methods to extend the neck. However, this investigation was not able to determine how to operate the neck extension, contraction, and stop. Therefore, we focused on the upward direction and determined the operation methods through the investigation using “Tsumori Control”. This study was approved by the research ethics committee of Kobe University (Permission number: 03-30) and was carried out according to the committee’s guidelines.

### 3.1. Assumed Environment

In this paper, we propose a human-augmentation system that expands the viewpoint movements by extending and contracting a virtual neck. It would be useful if we could freely extend and contract our necks in our daily lives. For example, when we look for someone in a crowd, we could easily find that person if our neck could extend and contract upwards. When we want to see a sign on the other side of a river, we can read the sign without going to the other side by stretching our neck forwards. Additionally, when we want to see a huge picture in front of us, we can see the whole picture if the neck is extended backwards.

Human movements depend on the distance of the object we want to see. As shown in Figure 3, when the object we want to see is on the other side of a wall that is taller than us, we thrust our chin up; when the text we want to see is far away, we move our head forward. In addition, if the text we want to see is too close, we pull our heads back. Therefore, the input methods of the system require intuitiveness and high responsiveness. We need to design an input method that is natural for humans. In order to intuitively extend and contract the virtual neck, the system needs functions that allow us to stop, extend, and contract the neck when we want. In addition, for daily use of the system, we should not use handheld devices such as a controller.

### 3.2. Investigation of Natural Gestures

We should use natural gestures as input methods for a human-augmentation system that can extend and contract a virtual neck. However, the human neck does not extend in reality. Therefore, we determined the gesture input to be the motion that appears when the user tries to extend his/her neck. This motion has not been investigated in previous studies. We set up three situations in which the participants tried to extend their neck upward, backward, and forward. We asked the participants what motions they would make when looking at the target in these situations, and they demonstrated them. The situations are (a) to (c) below.

(a)UpwardsThe participants looked beyond of the wall that is taller than their height. They read the text on the other side of the wall, which was about 5 cm higher than they were.(b)ForwardsThe participants looked far. They performed an vision test situation in which they read text 5 m away while their upper body was free and their legs fixed in place.(c)BackwardsThe participants looked at the large object nearby. We chose a large enough wall, 180 cm wide and 90.5 cm long.

If we are able to find common motions among the participants in these observations, we will employ these motions as input gestures. The participants were four randomly selected lab members in their twenties (three males and one female). In (a) to (c), they could move their upper body freely without changing the location of their feet.

Figure 4 shows the results of the investigation. This scene was judged to be an intentional movement of the participants who wanted to extend their necks in each of the movements from (a) to (c). In Figure 4a, they make “thrusting the chin upward”, “thrusting the chest out” and “standing on tiptoe” motions. These motions appeared because they tried to raise their viewpoint as much as possible to read the letters on the other side of wall. In Figure 4b, they make “thrusting the face forward”, “bending the waist forward” and “squinting the eyes” motions. They tried to get as close as possible to read the words. In Figure 4c, they make “pulling the head back” and “bending the waist” motions. They tried to pull their viewpoint back to read the large text in front of them. We judged these motions to be natural for humans, since they appeared to all participants.

### 3.3. Sensor Placement

From the observation shown in Figure 4, we found that the natural gestures for upward, forward, and backward stretching varied according to the neck and chin relationship. Therefore, we focused on the neck and the chin as input gestures for this research. The following input gestures were determined:(a)UpwardThrusting the chin upward(b)ForwardThrusting the face forward(c)backwardPulling back the neck

Figure 5 shows the sensors and their placement for recognizing these input gestures. As shown in Figure 5a, a Stretch Sensor [21] of IMAGES SCIENTIFIC INSTRUMENTS, which can acquire sensor values of elongation, was connected to a choker wrapped around the chin and neck to recognize chin thrust. As shown in Figure 5b,c, a Flex Sensor [22] of HIROTEC Corporation was placed at the front of and behind the neck to recognize the motion of moving the face forward and back. The reason for using these sensors is that they can acquire only the considered gestures, regardless of the user’s movement. In addition, as a result of our observation and research, we focused on the relationship between the chin and neck and thought it would be easy to attach a sensor to these parts to obtain their movements.

We placed the Flexible Stretch Sensor [21] of IMAGES SCIENTIFIC INSTRUMENTS, which can acquire sensor values for stretching, in front of the neck, and the Flex Sensor [22] of HIROTEC Corporation, which can acquire sensor values for bending, in front of and behind our neck. Thereby, we could obtain the sensor values of the chin movement and the back and forward angles of our neck. The reason for using these sensors is that they can acquire only the considered gestures, regardless of the user’s movement. In addition, as a result of the observation, we focused on the relationship between the chin and neck, and thought it would be easy to attach a sensor to these parts to obtain their movements.

In this paper, we implemented a prototype that supports neck extension and contraction with only upward movements. Therefore, we used a stretch sensor to connect the chin to the neck, and treated the thrusting motion of the chin as an input gesture. We were careful about how to attach the sensor so as not to cause discomfort to the users. Figure 5a shows the area around the sensor. We connected the choker to the frame of the mouth guard with a stretch sensor. The stretch sensor values were smoothed by a low-pass filter due to the noisy nature of the sensor values. The calculation formula is as follows: $X_{i}$ is the output value, $X_{i-1}$ is the previous output value, *S* is the stretch sensor value, and 0<a<1 is a constant.
$$X_{i}=\alpha *X_{i-1}+\left(1-\alpha \right)*S$$

### 3.4. Parameter Setting Using “Tsumori Control”

In Section 3.3, we decided to use a stretch sensor connected to the chin and neck to recognize the upward thrust of the chin, which is an input gesture for a human-augmentation system with an upward extending and contracting of the neck. In addition, we needed to decide the parameters such as the speed of the neck while it is extending/contracting and the method to stop the neck. To determine the operating parameters, we used “Tsumori control”, which allows the user to determine intuitive operating parameters based on the sensor input values when the user intends to operate a moving object [20].

We conducted experiments to determine the parameters that control the speed at which the neck is extending and contracting, and how to stop the neck. Figure 6a shows the experimental environment. We showed the participants videos of an avatar whose neck extended and contracted five times. The camera behind the avatar in the videos moves up and down as the avatar’s neck extends and contracts. The mirror in front of the avatar showed the participants a clear view of the avatar. We asked the participants to move as if they controlled the avatar with a stretch sensor connected to the chin and neck. We expected that the participants’ motion and the stretch sensor values would change when the neck stretching speed changed, so we prepared six different videos with different neck stretching speeds. As shown in Figure 6b, the neck of the avatar standing in front of the mirror stretches upward and downward in the videos. We prepared three types of constant neck stretch speeds, slow, normal, and fast, and three other types that slow down when the neck is fully extended at each speed. The participants were five male university students. They are in their twenties. One person participated in the observational study. There were no recruitment criteria. Others were chosen at random.

Figure 7 shows the stretch sensor value of one participant and the change in the head position of the avatar in the video. The results for all participants are shown in Appendix A. This tendency of the change in the stretch sensor value was the same for all participants. The results show that the participants intended to control the avatar so as to map the stretching of the avatar’s neck and the sensor. The participants moved to align their own viewpoint with the avatar’s up-and-down viewpoint. The delay between the change in the value of the stretch sensor and the change in the length of the avatar’s neck became smaller each time, which indicates that the participants intended to control the avatar with the stretch sensor.

Figure 8 shows the assignment between the stretch sensor value and the neck length. In the figure, the horizontal axis shows the stretch sensor value (*s*) and the vertical axis shows the virtual head position (*h*). We calculated the virtual head position (*h*) as
(1)$$h=\frac{\left(h_{2}-h_{1}\right)\left(s-s_{1}\right)}{s_{2}-s_{1}}+h_{1}$$
where $s_{1}$ is the stretch sensor value when the neck is not stretched ($h=h_{1}$) and $s_{2}$ is the stretch sensor value when the neck reaches the limit of full stretch ($h=h_{2}$).

### 3.5. Prototype System

In this study, we assumed a body-augmentation system to be used in an AR environment, and we constructed an environment where the system can be tested in a VR space that can simulate a variety of situations.

#### 3.5.1. Device and Software

As shown in Figure 9, we implemented the prototype in which the virtual neck extends upward in VR maze. We set up a sensor placement for recognizing input gestures, outlined in Section 3.3, and applied it to this system. Figure 10 shows the configuration of the prototype system. We used Oculus Rift S (Oculus [23]) as the HMD. Among the input gestures investigated in Section 3.2, the stretch sensor connected to the chin and the neck recognizes the thrusting the chin upward. The value of the stretch sensor is sent to the PC through the microcomputer, Arduino Nano (Arduino Holdings [24]). For the implementation of the VR maze, we used Unity 3D game engine (Unity Technologies [25]).

#### 3.5.2. Controlling the View Field Image

As shown in Section 3.4, we controlled the view field by assigning the value of the stretch sensor attached to the neck to the position of the head. We calibrated the sensor values for each user’s operation method. To calculate the linear equations for each user by means of Equation (Equation 1), s1 and s2 were obtained in the same way as experimental the environment of “Tsumori control” in Section 3.4. Then, the head position in VR space is controlled by the stretch sensor values.

#### 3.5.3. Application Example

We implemented a maze in the VR space. Since this maze is surrounded by walls higher than the user’s viewpoint in the VR space, the visibility is poor and it takes time to reach the goal. First, we connected the user’s chin and neck with a stretch sensor for upward input as investigated in Section 3.2. When the user wants to see the maze from a bird’s-eye view, they make the gesture that becomes the cue for the input. Then, the system recognizes it, and the user gets a bird’s eye view.

## 4. Evaluation

We confirmed the appropriateness of the input and control methods of the system for human augmentation that extends the neck upward, as discussed in Section 3. We compared the handheld controller with the proposed manipulation method in terms of how fast the neck extension/contraction can be positioned. Our target is the AR environment; however, we chose the VR environment as an evaluation environment since it allows us to test our methods in a variety of situations. We used the prototype system built in Section 3.5.3 as a simple test application without any movement during our evaluation experiments. In the evaluation experiment, the participants performed tasks using each operation method. We compared the time to complete tasks in each operation method to evaluate the operability.

The nine participants (two females and seven males) were recruited from the university and the general public. One participant participated in the previous investigations and experiments, and two participants participated in either the observation or the “Tsumori control” experiments. Four participants were new to the evaluation experiment. They ranged in age from 22 to 60 years old, with a mean of 30 years old (SD = 14). The reason for recruiting participants over a large age range was to identify the task completion times by age. The affinity for the immersive VR interface was slightly high (M = 3.0; SD = 1.5; measured on a 5-point Likert scale from 1 (unfamiliar) to 5 (skilled)). One of the participants (Participant A) took part in the experiment with the author as the skilled operator. Some participants used the immersive VR interface for the first time (n = 4), several times a year (n = 3), several times a month (n = 1), and several times a week (n = 1).

### 4.1. Experimental Method

#### 4.1.1. Experimental Environment

Figure 11 shows the scene of the participant during the experiment. The participants performed the experiment standing upright. Figure 12 shows the view field image presented on the participant’s HMD at the start of the evaluation experiment. The participants controlled the visual field image by the stretch sensor value described in Section 3.5.2. In the VR application for the evaluation, we set the virtual body that moves synchronously with the participant’s movements in real time. During the experiment, the participants held the Oculus Touch controllers shown in Figure 13 in both hands, which is capable of tracking both hands. The head movements were tracked by the HMD. Therefore, the participants’ hands and head movements were synchronized with the virtual body. The user could not grasp the virtual body since their viewpoint is the first-person viewpoint in the VR space. Thus, we set up a virtual mirror in front of the virtual body as shown in Figure 12. The mirror told the user the current state of the virtual body.

In the VR space, a sense of body ownership is not generated because one cannot see one’s own body. The sense of bodily extension is important because it relates to a sense of immersion and augmentation. We introduced a phase in which the virtual body is manipulated in front of the virtual mirror that synchronizes with the user’s movements, referring to the study of Tabitha et al. [14]. The phase induces illusions of ownership over the virtual body.

We also devised a way for the participants to obtain the sensation that their necks extend in the VR space. In the same way as the control method of the visual field image, the system plays a sound effect that gives the users the impression of an object being extended.

#### 4.1.2. Operation Method

We described how to operate the system with the handheld controller and with the gesture input. Throughout the entire task, the participants held the Oculus Touch controller shown in Figure 13 in both hands. The participants operated the system with the controller held in their left hand. The neck of the virtual body in the VR space extended when the user moved the left thumb stick forwards, and contracted when the user moved it backwards. The neck stopped extending and contracting when the user released the stick.

The natural gestures are the same as those described in Section 3.2. Figure 14 shows the gesture operation method. We used a stretch sensor connected to the chin and the users’ necks to recognize the upward thrusting of the chin. The extension and contraction of the virtual neck were controlled in the same way as in the visual field image described in Section 3.5.2. The neck extends upward when the user thrusts the chin upward, and the neck contracts when the user looks down. The neck stops extending and contracting when the user faces forward.

There are limits to both extension and contraction. The neck is not shorter than the original neck. The neck length is mapped to the degree of chin thrust, so the neck length reaches its upper limit when the chin is fully thrust. If the participant continues to tilt the VR controller stick upward, the neck extends all the way.

#### 4.1.3. Procedure of the Experiment

As shown in Figure 11, the participants wear the HMD and hold the handheld controller with both hands. Figure 12 shows the visual field image presented on the HMD. The virtual body, which is synchronized with the participants’ movements, is standing in front of the virtual mirror.

First, we instructed the participants to make simple movements of the upper body, such as shaking hands and head. By doing so, the virtual body reflected in the virtual mirror moves in synchronization with the participants, inducing illusions of ownership over the virtual body.

Secondly, participants performed the task a total of 16 times, eight times each, using a controller input and gesture input. In each method, we gave the participants time to get used to the system before the first task. As shown in Figure 15, the participants performed the task of aligning their heads by extending the virtual neck to the cube object above their heads in the VR space. Moreover, they kept the neck at the position of the object for three seconds. When the participants extended the neck to the object, the color of the object changed and the timer presented in the view field started. The task was completed when the color of the object changed for three seconds and at the same time the timer showed “clear!”. We recorded the time to complete the task and the y-coordinate of the virtual body’s head. We counted four tasks in each operation method as one set. The participants performed the task in a total of four sets (two sets of controller input and two sets of gestures) alternately. Five participants performed the task from the controller input set, and the others from the gesture set. In order to prevent the task from becoming too simple due to familiarity, we alternated the height of the objects (h=6,8).

The stretch sensor for recognizing gestures was attached before each gesture task and removed after each task. Before the first set of gestures, we calibrated the stretch sensor values using the “Tsumori control”. The calibration was performed by obtaining the two points $\left(s_{1},h_{1}\right)$ and $\left(s_{2},h_{2}\right)$ described in Section 3.4, and assigning a stretch sensor value and a virtual neck length to each participant.

After finishing each task with each operation method, participants filled in the questionnaire. Each of the following questions was asked: “Did you feel your neck extending and contracting?”, “Did you feel fatigue?”, “Did you feel video sickness?”, “Did you feel immersed?”, “Did you feel the operation was easy?” and “Did you feel you could operate the system as you wanted?”. The questionnaire is a five-point Likert scale with 1 representing “Strongly disagree”, 3 “Undecided /Neutral” and 5 “Strongly agree” to the questions. In addition, the participants responded to the system with open-ended comments.

### 4.2. Result

#### 4.2.1. Results on Task Completion Time

Figure 16 shows the average time taken by the nine participants to complete the task with each operation method. The vertical axis of the graph is the elapsed time from the start of the virtual neck extension to the completion of the task, and the horizontal axis is each operation method. The error bars show the standard deviation.

We performed a within-participant two-factor analysis of variance on the task completion time results, and the results showed a significant differences between the methods (F(1,8)=32.18,p<0.01), as shown in Figure 17. The time to complete the task was shorter with the controller input than with the gesture input. There were two participants for whom the results of the second set of task completion times for gesture manipulation were much better than the first set. This is due to the habituation of the task. In addition, the results of the skilled gesture operators show they completed the task with the same operability as the controller, and the results from the gesture operation of the elderly (Participant E, 60 years old) do not lose to the controller by a large margin. The results of this experiment show that gesture operation can have the same operability as the controller with a practice in a wide age range. The gesture operation, which is not too different from the controller shown in this experiment, can be used in daily life since the hand-free operation is a big advantage in the assumed daily life scenario.

On the other hand, we need to improve the gesture operation. Figure 18 shows the transition of the head position during the task as a characteristic of each operation method. The vertical axis of the graph shows the y-coordinate in VR space, and the horizontal axis shows the time passed from the start of the virtual neck extension. The participants performed the extension of the neck to the object at the height y=6. Figure 18a shows that the participants sometimes overshot the target by gesture operation. After extending the neck a little past y=6, the user may contract the neck too much or extend it too vigorously. The reason why it took them longer to complete the task with the gesture input is that we have not yet established the way to stop the neck or fine-tune the extended neck. We may need to add other sensors for the adjustment. In contrast, Figure 18b shows that the controller input was able to finely adjust the neck length to the target point.

#### 4.2.2. Results of Questionnaire

Figure 19 shows the results of the questionnaire: one indicates “Strongly disagree”, three indicates “Undecided/Neutral” and five indicates “Strongly Agree”. We converted this subjective evaluation into a score. In other words, a higher score indicates agreement with each question. We performed t-tests on the gesture and controller scores for each question item. As a result, there was a significant difference between the two operating methods for questions Q1 (t(8)=6.33,p<0.01), Q2 (t(8)=3.29,p<0.05), Q4 (t(8=2.4648,p<0.05)), Q5 (t(8)=4.00,p<0.01) and Q6 (t(8)=5.98,p<0.01) as Figure 19 shows. Therefore, the gestures extracted in this study were significantly more neck extensible, immersive, and fatiguing than the controller. On the other hand, the controller was significantly easier to operate than the gestures. The fatigue felt by some participants may have been caused by repeatedly moving their chin up and down when they had difficulty positioning their heads.

We obtained the following comments from the participants.

The controller input reduced my immersion as I repeated the task, because the task gradually became simple thumbstick movements. On the other hand, the gesture input did not decrease my immersion. Because the body movements linked to the viewpoint movements, and I need to move my body to operate this system.The virtual neck contracted when I tried to look at the target object in front of their faces.When I operated the virtual neck with gestures, I needed to raise my face so that I could grasp the distance to the target object.

These results suggest that human augmentation with hand controllers is suitable for tasks such as work sites. For example, we considered that human augmentation with a hand controller would be suitable for searching for occlusion in a confined space. In fact, the participants in the controller set were simply moving their thumb sticks without looking at what was above them, as if they were working. However, we believe that the proposed method is effective in everyday scenarios because it can be operated hands-free. In addition, we believe immersion and a sense of augmentation are important for use in everyday life and gestures make it easier for people of all ages learn how to use human body augments. The gestures extracted in this study were found to be accompanied by a sense of extension, but since no comparison was made with other gestures, it is difficult to say for certain that the gestures in this study are superior. System improvements are needed to improve operability and compare with other gestures.

## 5. Limitation

In this paper, we described that the target of our system is AR environments. However, in our evaluation, we tested the prototype in a VR environment to perform a flexible evaluation. We have several remaining tasks to apply our system to AR environments:The stretch sensor used in the evaluation experiment was noisy and required smoothing. This caused a delay.The number and variety of participants were limited.It is necessary to consider how to distinguish between the input gestures investigated in this study and other actions performed in daily life.

The details of each are discussed below.

### 5.1. Sensor Limitations

The reason why the participants could not fine-tune the neck as they wanted may be because the stretch sensor used in this study had a lot of noise in the sensor value and there was a delay caused by the smoothing process used to control the stretch sensor. In this paper, we proposed a method in which the stretch sensor recognizes the chin thrusting motion and assigns the sensor value to the neck length. Another method is to use the angle of the head for control. It may be possible to obtain a more accurate control by using an acceleration sensor instead of the stretch sensor to obtain the angle of the head.

In addition, there was a case in which the neck contracted contrary to the intention to stop the head. Since the proposed method assigns the stretch sensor value to the length of the neck, the neck stops stretching when the degree of stretching of the stretch sensor is maintained. Therefore, when the participants stretched his/her neck to the target object, he/she tried to look at the target object in front of his/her face, which caused the stretch sensor to loosen and the neck to contract. Therefore, as a new control method to stop neck extension and contraction, we consider a way to stop neck extension and contraction once the stretch sensor is loosened after a gesture input.

### 5.2. Participant Limitations

In each experiment, there was an overlap of a small number of participants. In the evaluation experiment, all participants were given practice time first, so we think that the effect of this overlap is small. However, further investigation is required because priming and expectations possibly affect the generalization of the results. In addition, there were few participants in all surveys and experiments. Including more participants of different age groups and backgrounds is important to generalize the findings of this study. Increasing the variety of participants is one of the future tasks.

### 5.3. System Limitations

In this paper, we observed the motions that people make when they want to stretch their necks, and examined natural gesture input methods. We decided to use a single stretch sensor to recognize the upward gesture input method of chin thrusting. In the evaluation experiment, we confirmed that skilled operators can intuitively operate the system. However, in our daily lives, we also thrust our chin up when we do not want to extend our neck. Therefore, it is necessary to distinguish between the input gesture for neck extension/contraction and other actions, such as chin thrust, when we want to extend our neck and chin thrust when we simply want to look upward. As shown in Figure 20, we thought that by the addition of the gaze information to the chin-up motion helps to distinguish between neck extension and other motions. In the same way, it may be possible to distinguish between the movements to contract the neck and the movements to look down.

In addition, the proposed system is intended for use in daily life, and for this purpose, a high responsiveness is required in addition an accurate operation. In the evaluation experiment, we confirmed the accuracy of the proposed method. In the future, we plan to implement the proposed method by combining acceleration sensors and eye tracking information, evaluate it in a task that requires responsiveness, and propose a practical operation method.

## 6. Conclusions

In this paper, we proposed a natural gesture input for a system for human augmentation that can extend and contract a virtual neck. We investigated natural gestures for the system input by observing human motions in a visually limited situation. From the results of the investigation, we decided on three gesture inputs: “thrusting the chin upward,” “thrusting the face forward” and “pulling back the neck”. In addition, we investigated detailed operation methods for a human-augmentation system that extends the neck upward using “Tsumori control”. From the results of the investigation, we controlled this system by mapping the stretch sensor values to the virtual neck length with a linear function. Then, we implemented a prototype system that extends the virtual neck upward in VR environment. We compared the conventional controller input method with the gesture input using the proposed system for evaluation. We found that the controller input was better in terms of speed. However, skilled gesture operators performed head positioning with the same performance as the controller. As a result of the questionnaire survey, the gesture input increased the sense of immersion and augmentation.

## Figures and Tables

**Figure 1 sensors-22-03559-f001:**
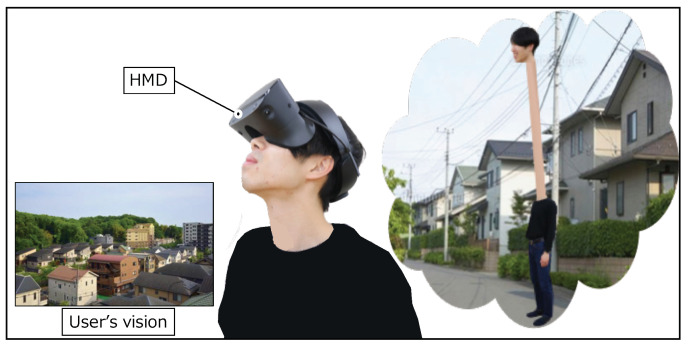
Extensible neck.

**Figure 2 sensors-22-03559-f002:**
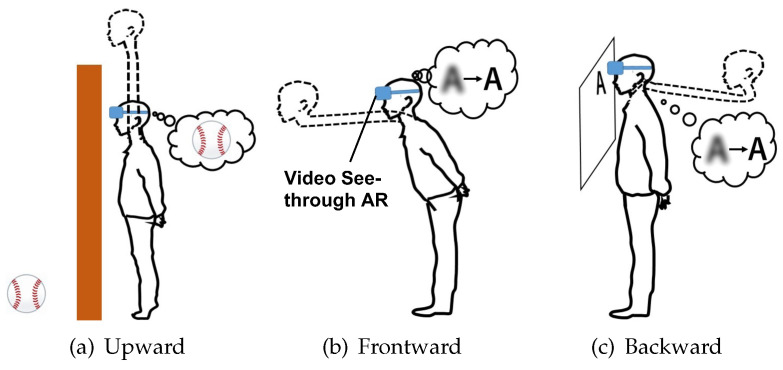
The situations using proposed system.

**Figure 3 sensors-22-03559-f003:**
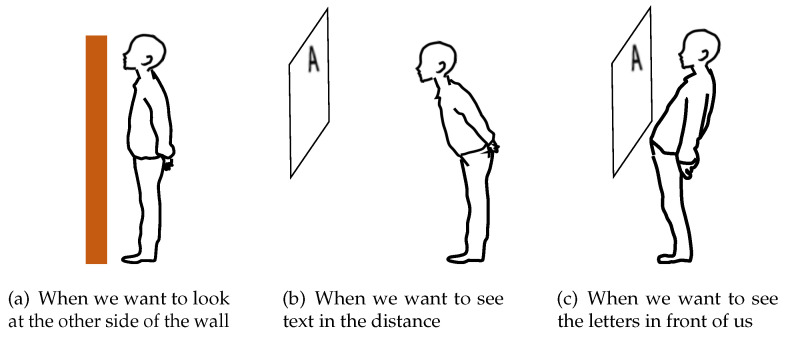
Motion in daily life.

**Figure 4 sensors-22-03559-f004:**
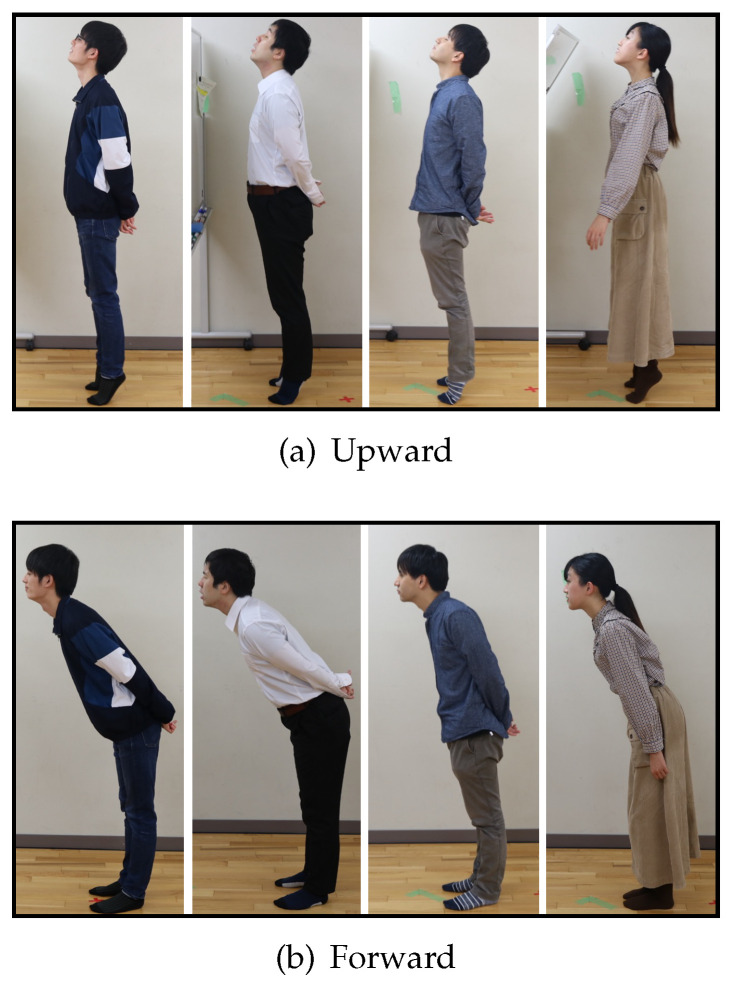
Participants’ motion.

**Figure 5 sensors-22-03559-f005:**
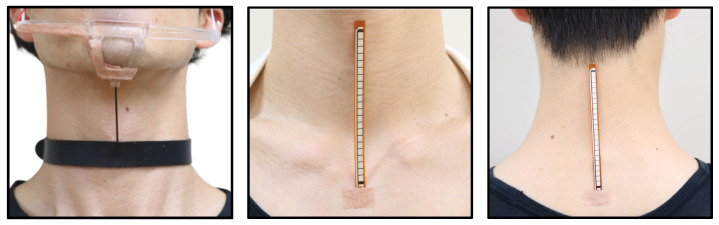
Sensor placements for recognizing each intuitive motion.

**Figure 6 sensors-22-03559-f006:**
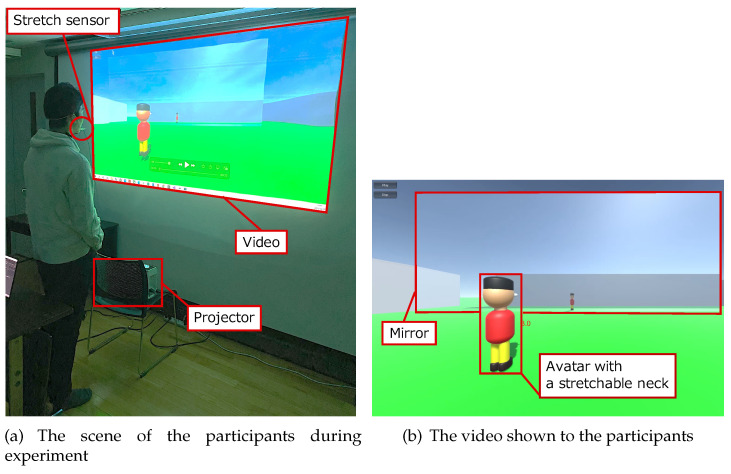
Experimental environment.

**Figure 7 sensors-22-03559-f007:**
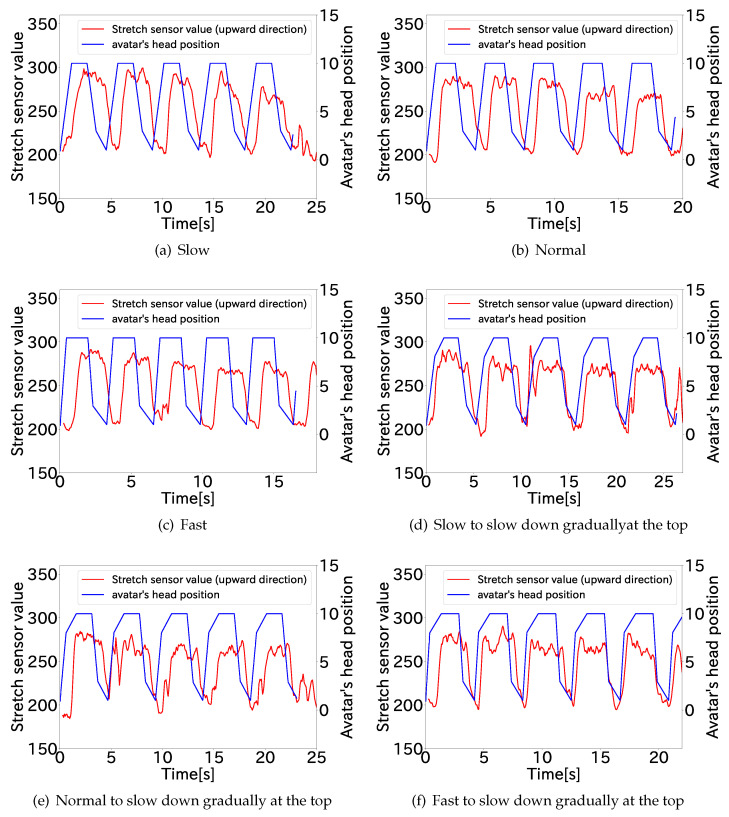
Stretch sensor values at each stretch speed in the upward direction (Participant A).

**Figure 8 sensors-22-03559-f008:**
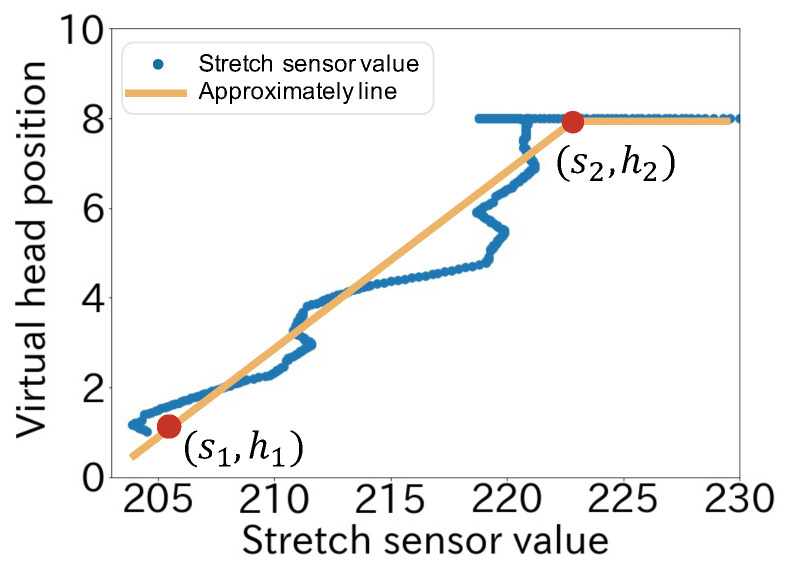
Mapping method in the upward direction.

**Figure 9 sensors-22-03559-f009:**
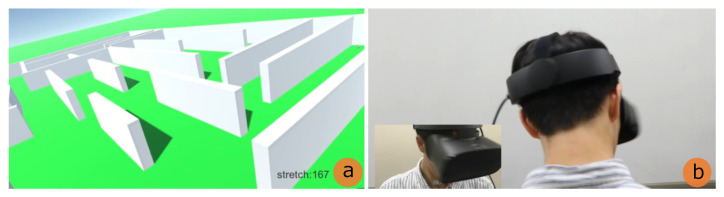
VR maze (**a**: user’s view field; **b**: user) (the user’s view is first-person perspective).

**Figure 10 sensors-22-03559-f010:**
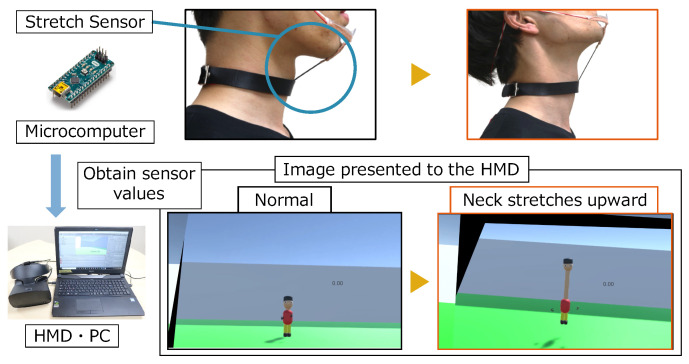
System configuration (the user’s view is first-person perspective).

**Figure 11 sensors-22-03559-f011:**
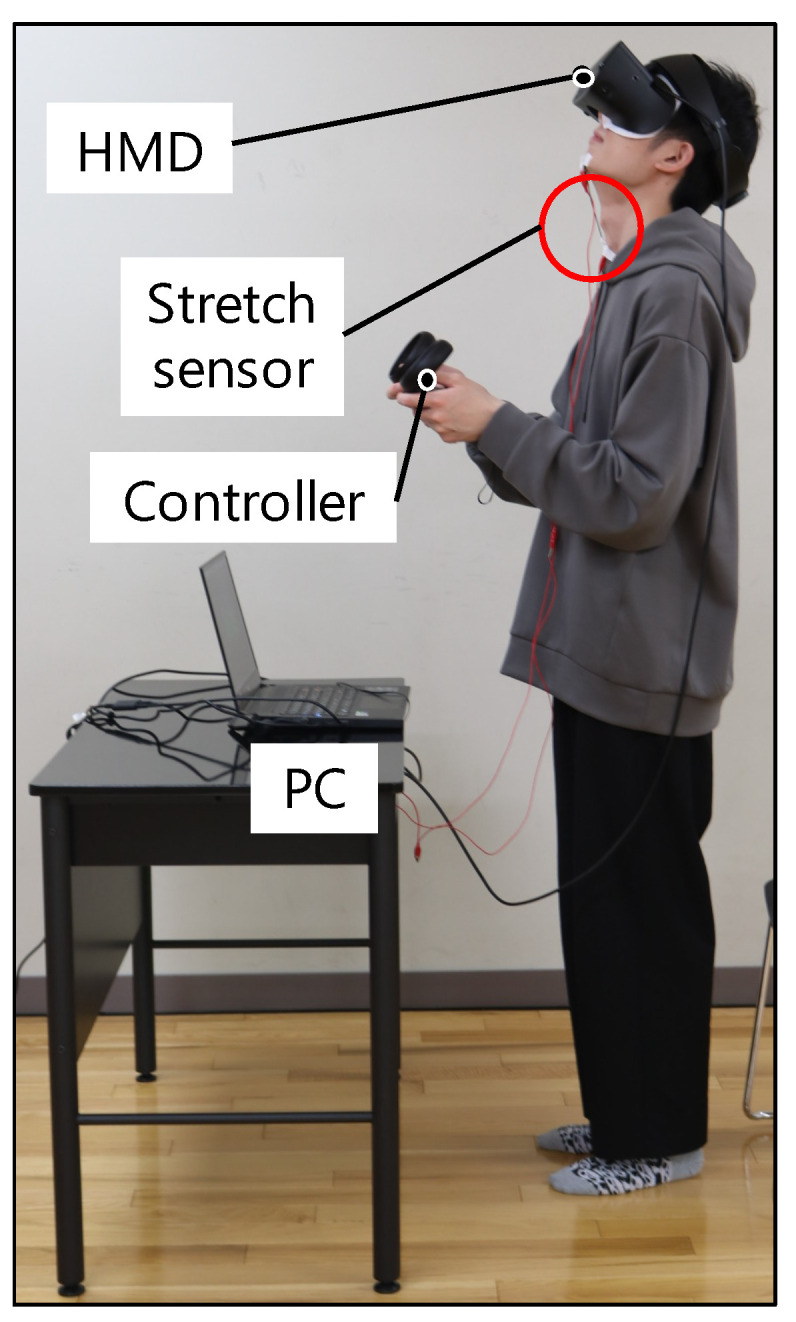
The setup of the experiment.

**Figure 12 sensors-22-03559-f012:**
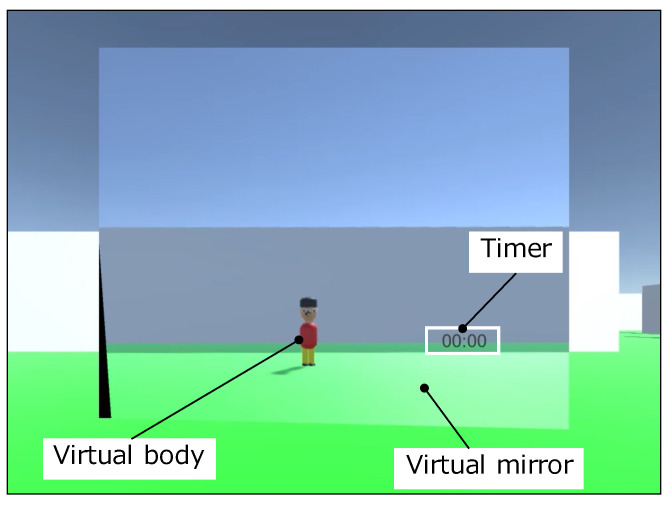
The visual field image presented on the HMD.

**Figure 13 sensors-22-03559-f013:**
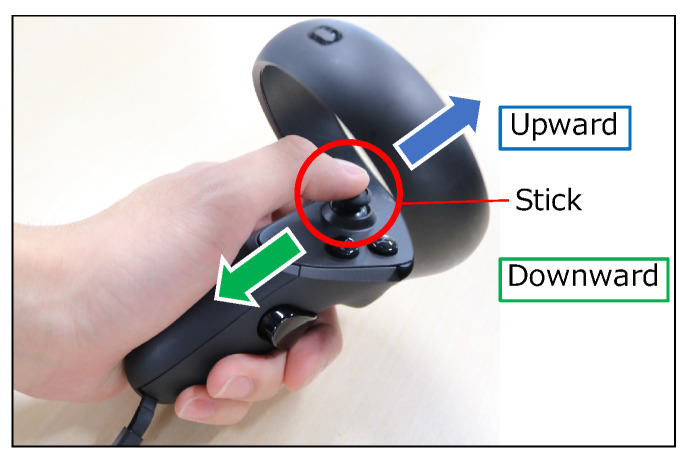
Operation with the handheld controller.

**Figure 14 sensors-22-03559-f014:**
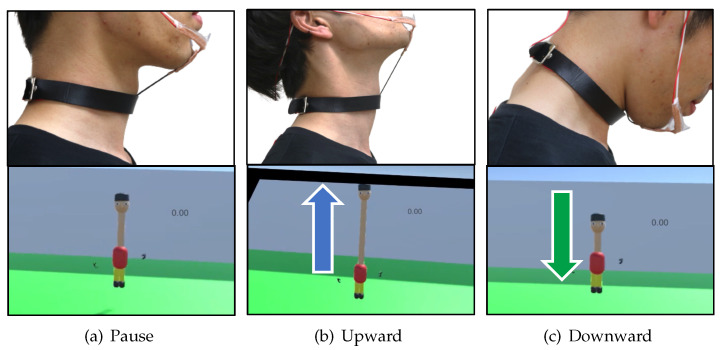
Operation with a natural gesture.

**Figure 15 sensors-22-03559-f015:**
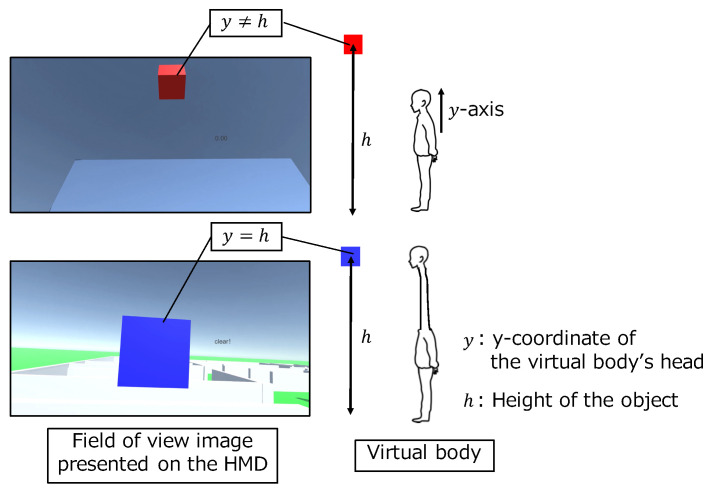
Task performed in the evaluation experiment.

**Figure 16 sensors-22-03559-f016:**
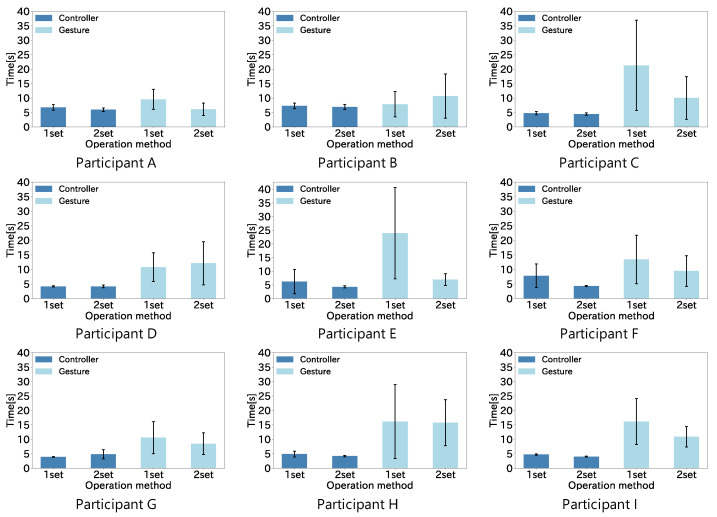
Task completion results for each operation method.

**Figure 17 sensors-22-03559-f017:**
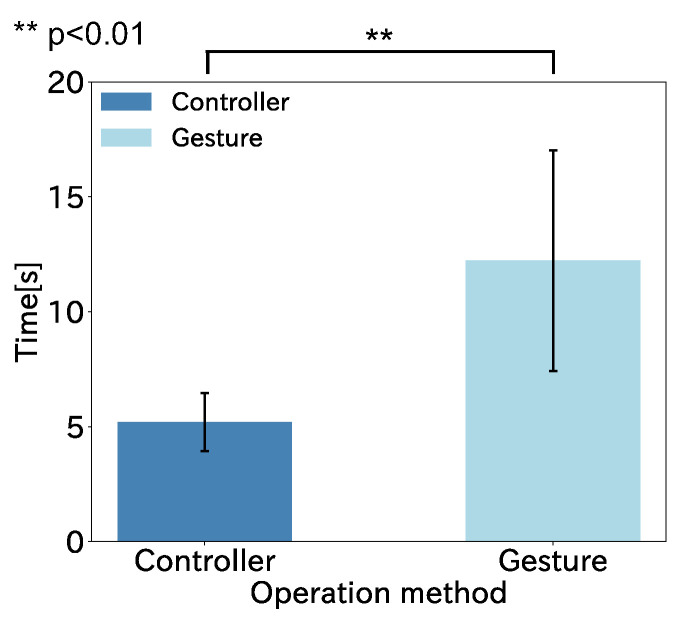
Task completion results summarized for all participants.

**Figure 18 sensors-22-03559-f018:**
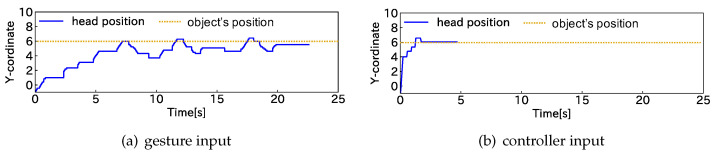
Transition of head position for each operation method during the task.

**Figure 19 sensors-22-03559-f019:**
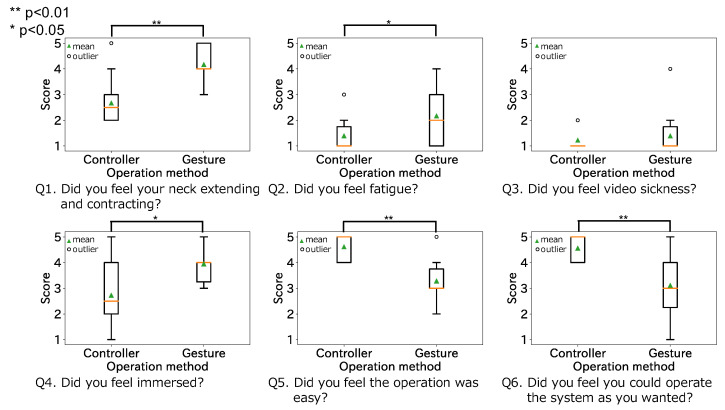
Results of the questionnaire.

**Figure 20 sensors-22-03559-f020:**
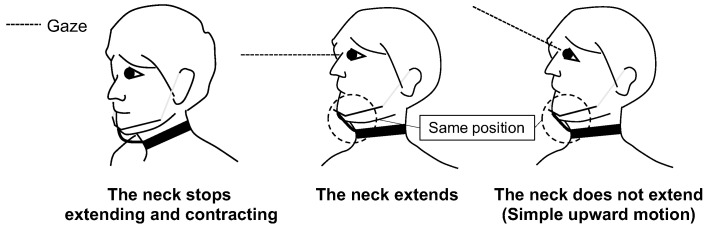
Diagram showing how to distinguish between daily motions and gesture input.

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
