# Peer review of "Extensible Neck: A Gesture Input Method to Extend/Contract Neck Virtually in Video See-through AR Environment"

_sensors, 2022, doi:10.3390/s22093559_

Round 1

Reviewer 1 Report

My comments are in the attached document.

Reviewer 2 Report

The authors present the article on a very important and interesting topic related to communication in the AR environment. At the beginning, the authors introduce the reader to the technology, documenting it with relevant and suggestive photos. This is an undoubted advantage of this part of the article that facilitates the understanding of the problem posed. They refer to appropriate sources which confirms a good problem analysis and knowledge of the state of art.
Then in the article you can find the system design and well-described research procedures. The drawings and illustrations used again bring the problem closer and make it possible to understand the problems and solutions presented by the authors. The prototype presented looks promising, as do the results that have been extensively commented on in the body of the article. By analyzing the results obtained by the authors, it can be presumed that the obtained results are a good introduction to further work and look promising in terms of commercial implementation. The article is worth publishing.

  1. The topic is original because the issue described by the authors can also be successfully implemented in commercial applications. Why? This results directly from the increasing use of the technologies in question, not only in games, but also in virtual trainers based on virtual reality.
  2. There is a completely different approach in the cited publications. The authors present a specific Extensible Neck: A Gesture Input Method - this proposal is promising as presented by the authors.
  3. Simply describe the methodology in more detail in order to increase the possibility of reproducing the study or to fully show possible threats.
  4. The conclusions relate to the study as much as possible, summarizing the results achieved in an atomic way. There are conclusions for this.
  5.  The references are appropriate, although of course you can always do more.
  6.  The illustrations shown are very accurate. Both those showing the examination and showing in detail, for example, the movements of Fig.15, but also the earlier ones. The shown stretch sensor value for each stretch sensor speed, both in the main part of the publication and in additional materials, give a good knowledge of the measurement results. The earlier illustrations clearly illustrate the measurement itself, the problem and the obtained results, as in Fig. 18. 

Reviewer 3 Report

This paper presents a system that expands the movements of the user's viewpoint by extending and contracting the neck in a VR environment. They implemented a prototype by using Oculus Rift for the human-augmentation system to control the virtual neck. It is an interesting work and has some potential useful scenarios. The entire system is quite complete. 

There are some minor comments:

  1. Since the developed prototype is designed on a VR system, it is more suitable to replace the "AR Environment" into "VR Environment' in the title;
  2. The number of the participants in the case study is quite small, e.g., 9 participants. It is better to include more participants with different age groups, different backgrounds, etc. So that it will make the evaluation more general and extensible. 

Reviewer 4 Report

The authors proposed a method of assigning the degree of chin thrust to a virtual neck length through an investigation using “Tsumori control [5],” in which behavioral intentions can be assigned to intuitive control methods. Experimental results show that the proposed method appears to be effective. Detailed comments are listed as follows,

The contributions of this works should be explicitly given in the section of introduction.

Is the training way of the proposed method end-to-end? More details of the network configuration should be given.

The title of this paper should be changed. The current title could not reveal the novelty of this work.

Some works also explore the human body analysis, such as

Spatiotemporal co-attention recurrent neural networks for human-skeleton motion prediction. IEEE Transactions on Pattern Analysis and Machine Intelligence (TPAMI). 2021 

Skip-Attention Encoder-Decoder Framework for Human Motion Prediction. Multimedia Systems, 2021.

The authors should introduce these works in the revision.

Round 2

Reviewer 1 Report

I thank the authors for their answers to my comments. Some final suggestions follow.

Revised section 4.1 states some differences in the time between controller and gesture after statistical analysis. However, figure 16 does not show what the statistical difference was.

The fact that some participants participated in more than one experiment could mean that there was a priming, bias or expectation towards the experiment, and this could affect the generalization of the results.

Authors responded the question about Figures 7-8 but the figures were not adjusted. In addition, I still believe that, in general, the Figures’ legends in this paper should be improved. The authors may think about readers that may want to scan the manuscript first by looking at the figures (and legends), that should be self-explanatory and the reader, based on the figures, should be able to have a basic understanding of the whole study.

The same as above go to the questions regarding figures 9-10.

I suggest including the answer to “Point 30” (from the answers to my comments) in Figure 19

Review English. For instance, the following sentences, including at least one that I mentioned already in the first review and was not corrected:

  • Based on this hypothesize, we constructed a system and evaluated the input interface in this paper.
  • “no research has been investigated what kind of motions are intuitive to people”. I believe the authors want to say: no research has investigated….Without the “been”. This is only one example, I spotted more English mistakes like these
